# Test and Study of Pipe Pile Penetration in Cohesive Soil Using FBG Sensing Technology

**DOI:** 10.3390/s20071934

**Published:** 2020-03-30

**Authors:** Yonghong Wang, Xueying Liu, Mingyi Zhang, Songkui Sang, Xiaoyu Bai

**Affiliations:** Institute of civil engineering, Qingdao university of technology, Qingdao 266000, China; hong7986@163.com (Y.W.); liuxueying@hnu.edu.cn (X.L.); 18306426194@163.com (S.S.); baixiaoyu@qut.edu.cn (X.B.)

**Keywords:** Fiber Bragg Grating sensor, jacked pile, double-wall open-closed pipe pile, earth plug effect, pile driving resistance, compaction effect

## Abstract

In order to examine the applicability of Fiber Bragg Grating (FBG) sensing technology in the static penetration of pipe piles, static penetration tests in clay were conducted using double-wall open and closed model pipe piles. The strain was measured using FBG sensors, and the plug height was measured using a cable displacement sensor. Using one open pile and two closed piles, the difference in pipe pile penetration was compared and analyzed. Based on FBG sensing technology and the strain data, the penetration characteristics of the pipe pile, such as axial force, lateral friction, and driving resistance were examined. Results showed that FBG sensing technology has superior testing performance for the pipe pile penetration process, can accurately reflect the strain time history of pipe piles, and can clearly reflect the penetration process of pipe piles with increasing penetration depth. In addition, the variation law of the characteristics of the jacked pile pile–soil interface was obtained. This test has significance for model tests and the engineering design of pipe piles.

## 1. Introduction

Jacked pile is widely used in China [1,2,3,4]. Jacked piles are a significant component of many construction projects; hence, improving jacked pile technology is an urgent need. However, many aspects of the current research are not sufficiently in-depth [5,6,7,8]. The research on the penetration mechanism of jacked pile in viscous soil foundation is relatively small, and while its penetration mechanism is affected by many factors, it is difficult to find an effective research method. In recent years, many domestic and foreign scholars have examined temperature compensation technology, packaging technology, installation technology, and engineering applications of FBG sensors: Nellen et al. [9] first measured the strain attenuation in a cable anchorage head during cable loading and operation using embedding sensors, conducted mechanical and thermal load tests, and calibrated the temperature and strain sensitivity. The reliability of wavelength shift caused by stress transfer, optical fiber mechanical failure, and thermal FBG decay was introduced. Albert and Shao [10] presented a relatively recent variant of the fiber grating concept, whereby a small tilt of the grating fringes causes coupling of the optical power from the core mode to a multitude of cladding modes, each with its own wavevector and mode field shape. Buerck et al. [11] proposed a distributed polymer-clad quartz fiber sensing system based on optical time-domain reflectometers (OTDR). Li et al. [12] reviewed fiber optical sensor health monitoring in various key civil structures, including buildings, piles, bridges, pipelines, tunnels, and dams. Chan et al. [13] introduced, investigated, and developed the feasibility of fiber Bragg grating sensors for structural health monitoring. They compared the performance of the fiber Bragg grating sensor with traditional structural health monitoring systems by monitoring the strain on different parts of the third tier under railway and highway loads. Majumder et al. [14] reviewed the latest research and development activities for structural health monitoring using fiber Bragg grating sensors and highlighted areas for further research. Barrias et al. [15] discussed and summarized the possibilities of using existing fiber optic sensors in structural health monitoring (SHM) systems to help understand and monitor the distributed behavior of key structures. Ren et al. [16] proposed an indirect pipeline corrosion monitoring method based on circumferential strain measurement and introduced a mechanism for enhancing circumferential strain measurement sensitivity and manufacturing technology. Its reliability and dynamic characteristics were discussed.

Currently, some scholars are adopting FBG testing technology in the jacked pipe pile field test. Baldwin et al. [17] embedded fiber Bragg grating sensors in the winding composite pipe of the bearing pile to obtain structural health monitoring data. FBG technology can be used to monitor pile health, including pre-installation, installation, residual strain during installation, and pile life. Lee et al. [18] evaluated the applicability of fiber-optic sensor systems for pile foundation detection based on a series of indoor and field tests. Kister et al. [19] monitored changes in strain distribution due to the simultaneous action of floor construction and ground uplift by installing Bragg gratings in reinforced concrete piles. Li et al. [20] used fiber grating sensing technology to monitor the strain field of PHC (high strength prestressed concrete) pipe piles during pile jacking. After the hydraulic pressure is removed, the residual strain of the pile body was obtained. The applicability of the fiber grating sensor in monitoring the strain distribution of the pile during jacking had been demonstrated. Gao et al. [21] monitored the deformation of PCC (Large diameter pipe pile by using cast-in-place concrete) piles based on a new quasi-distributed fiber sensing technology while the strain of PCC model piles based on FBG sensors was monitored. However, the traditional monitoring technology used during the pile jacking process of indoor model pipe piles is primarily a strain gauge [22,23,24]. Due to the small size of the model pipe pile and the small pressure on the pile, the size and sensitivity of the sensor were required to increase, and electrical sensors could not meet the test requirements. Fiber Bragg grating sensors have been applied to the model test due to their strong anti- electromagnetic interference ability, small size, light weight, and remote measurement [25,26,27,28,29,30,31,32,33].

Using FBG technology, this contribution examines the penetration characteristics of pile axial force, pile lateral friction resistance, pile driving resistance, and the soil plug effect during pile jacking. The applicability and accuracy of FBG sensors used in the penetration test of jacked piles are explored, providing reference and scientific basis for the model test and engineering design of jacked piles.

## 2. FBG Sensor Working Principle

FBG is a grating, whose refractive index changes periodically. Light satisfying the wavelength condition is reflected back and the rest of the light passes through the grating (Figure 1). The alternating shading of the fiber core represents the periodic change in refractive index, and the wave length reflection condition is as follows [34]:

(1)$$\lambda_{b}=2n_{eff}\Lambda$$
where, $\lambda_{b}$ is the fiber grating central wavelength; $n_{eff}$ is the effective refractive index of the optical fiber core; Λ is the grating period.

The fiber grating period and effective refractive index are affected by strain and temperature, which lead to changes in the FBG central wavelength. The photoelastic effect also leads to changes in the FBG refractive index; therefore, both the measured strain and temperature cause variations in the FBG central wavelength [35]:

(2)$$\frac{\Delta \lambda_{b}}{\lambda_{b}}=(1-p_{e})\Delta \varepsilon +(\alpha_{f}+\xi )\Delta T$$
where, $\Delta \lambda_{b}$ is the change in the fiber grating center wavelength; $\lambda_{b}$ is the fiber grating center wavelength; Δε is the change in strain; ΔT is the temperature change; $p_{e}$ is the effective elastic coefficient of the grating; $\alpha_{f}$ and ξ are the grating thermal expansion coefficient and grating thermal light coefficient, respectively.

In this experiment, a negative expansion material was introduced into the package of the sensitized miniature FBG sensor. When the temperature increases, the wavelength shift of the internal grating can be offset by the negative expansion material, so the response to temperature changes is greatly reduced, and the influence of temperature changes in the laboratory test can be ignored. Therefore, when the external temperature change is not considered [36]:


(3)$$\frac{\Delta \lambda }{\lambda }=\left\{1-\frac{n^{2}}{2}\left[p_{12}-\upsilon \left(p_{11}+p_{12}\right)\right]\right\}\Delta \varepsilon$$


Where: Δλ is the fiber grating wavelength change; $p_{11}$ and $p_{12}$ are photoelastic effect constants; υ is Poisson’s ratio; Δε is the change in axial strain of the fiber.

Let $K_{\varepsilon }=\left\{1-\frac{n^{2}}{2}\left[p_{12}-\upsilon (p_{11}+p_{12})\right]\right\}\lambda$ to obtain [37]: 

(4)$$\Delta \lambda =K_{\varepsilon }\cdot \varepsilon$$
where, $K_{\varepsilon }$ is the fiber grating strain sensitivity coefficient.

## 3. Test Scheme

### 3.1. Test Equipment

The test was conducted in a large-scale model box with a length of 3000 mm, a width of 3000 mm, and a height of 2000 mm (Figure 2). Above the model box was situated the loading system, and the loading hydraulic jack was placed on the loading system beam. The forward and backward movement of the hydraulic jack was controlled by a beam and electric control system, while the left and right movement of the hydraulic jack was directly controlled by the electrical control system. The jacking process was controlled by a hydraulic jack and RS-JYB static load test equipment. The whole test process is shown in Figure 3.

### 3.2. Model Pile

The double-wall open and end model pipe pile was selected to create friction between the outer and inner walls of the pipe pile. In order to prevent damage to the outer tube sensor during the pile jacking process, the two sides of the outer tube surface were slotted symmetrically, with a width of 2 mm and a depth of 2 mm. Two 5 mm-diameter inlet holes were installed in the outer tube, and 30 mm-diameter outlet holes were installed in the upper part of the outer and inner tube. The parameters of the inner and outer pipe of the model pipe pile are listed in Table 1.

Three model piles were tested. Pile ends were divided into closed and open piles, of which TP1 was an open pile, and TP2 and TP3 were closed piles. FBG strain sensors were installed on all three piles.

### 3.3. Strain Testing Technology

During the penetration process, the strain change of the model pipe pile was small, and the inner and outer pipe walls were thin, so test data were significantly affected by the external environment. A sensitized miniature FBG strain sensor was used for data collection in this experiment (Figure 4). The technical parameters of the sensitized miniature FBG sensor are listed in Table 2.

In this test, the FBG sensor was arranged along the length of the outer tube and the inner tube (Figure 5). The outer and inner tube sensor was symmetrically pasted to both sides of the tube and covered with 704 adhesive to protect the sensor. Twelve and 14 sensors were arranged symmetrically along the length of the outer and the inner tubes, respectively. The outer tube sensors were arranged with equal spacing of 180 mm, and the sensor spacing at the bottom of the inner tube was reduced. The stress mechanism of open pile and closed pile is shown in Figure 6.

### 3.4. Temperature Self-Compensating Pressure Sensor

In order to obtain the pile jacking resistance change law during pile jacking, a temperature self- compensation pressure sensor was installed on the top of the model pile to measure the pile jacking force. The pressure sensor had a diameter of 70 mm, a height of 25 mm, and a measurement range of 1 MPa, meeting the test requirements (Figure 7). The sensor was easy to install and did not need to be closely connected to the model pile. The sensor only needed to be placed horizontally on the top of the pile before starting to press the pile. The acquisition equipment for this sensor also used a fiber Bragg grating demodulator, which reduced the variety of acquisition equipment.

### 3.5. Data Acquisition System

The strain test adopted the FS 2200RM-Rack-Mountable Bragg Meter demodulation instrument (Table 3). The YWD-100 displacement sensor dynamically monitored the vertical ground displacement around the pile, and the DH3816N static strain tester collected the data from the YWD- 100 displacement sensor (Figure 8). The upper end of the pulley was connected to the MPS cable displacement sensor. The MPS cable displacement sensor recorded the pile settlement and height of the earth plug in real time. An XSR21R paperless recorder was used to record the data measured by the cable displacement sensor.

## 4. Analysis of Test Results

### 4.1. Development Rule of Soil Plug Height

The variation curves of plug height and burial depth of open model pile TP1 with penetration depth are shown in Figure 9. The model pile TP1 can be considered to be uniform penetration, and the earth plug effect gradually formed with increasing burial depth. At the beginning of the penetration stage, the increase rate of the earth plug height was faster and was similar to the penetration rate. As the burial depth continued to increase, the increasing speed of the plug height gradually decreased. Figure 9b shows the curve of soil plug height vs. burial depth, and the dotted line represents the soil plug filling completely. The actual soil plug height of the model pile was small and the soil plug height was 329 mm when the burial depth was 900 mm.

The occlusive degree of the plug was expressed by Increased Frequency Reporting (*IFR*), which was defined as follows [38]:(5)$$IFR=\frac{\Delta L}{\Delta D}\times 100\%$$

Figure 10 shows the curve of the *IFR* value of the earth plug with pile depth. In the initial penetration stage, the plug height increased rapidly with increasing burial depth. When the burial depth was about 200 mm, the *IFR* value decreased from 92.6% to 66.8%. As the burial depth continued to increase, the *IFR* value gradually decreased and the degree of soil blockage gradually increased.

### 4.2. Axial Force of Pile Body

By analyzing the FBG strain sensor data from the pile body as burial depth increased during pile jacking, the variation law of axial force on the pile body during pile jacking was obtained (Figure 11). The axial force of the outer pipe of the model pile gradually decreased from top to bottom with increasing soil penetration depth due to the play of lateral friction resistance in the pile. The axial force on the pile was greatest at the pile top, and the axial force on the pile body gradually decreased to the minimum value at the pile end. The axial force attenuation rate of the pile body changed with increasing burial depth. When burial depth was small, the lateral friction resistance of the pile was weak, and the axial force attenuation rate of the pile body was small. As the burial depth continued to increase, the lateral friction of the pile gradually increased, which was much higher than the lateral friction on the pile when the pile was initially buried; thus, the axial force attenuation rate of the pile increased. The friction resistance between the inner wall of the model pile TP1 and the soil resulted in an axial force on the inner pipe. The height of the plug increased gradually with increasing burial depth, and the formed plug tended to be dense. Therefore, the axial force on the tube in the same position increased continuously. With increasing burial depth, the friction resistance inside the pile gradually increased and the axial force of the inner tube decreased gradually with increasing soil depth.

### 4.3. Average Pile Side Friction Resistance

Average pile side friction resistance in the process of pile jacking τ*_av_* is defined as the pile side friction resistance buried area of pile body [39]:*P_s_* = *A_shift_* · τ*_av_*(6)
where, *A_shift_* is the buried area of the pile body, and *A_shift_* = π*dh*; τ*_av_* is the average pile side friction resistance. According to the total friction resistance of the pile side *P_s_* and the burial depth *h*, the average friction resistance of the pile side can be calculated using Equation (6).

The distance from the pile side FBG strain sensor to the pile end and the axial force on the pile body varied. Taking the closed model pile TP2 as an example, three sensors (*h*/B = 1, 3 and 5) were placed at different distances from the pile end on the pile side. *h* is the distance between the FBG sensor and pile end; B is the diameter of the model pile, corresponding to section A, section B and section C of the pile body respectively (Figure 12). Since the maximum burial depth of model pile TP2 was 900 mm, in order to ensure the reliability of the test data, only the sensors at the *h*/B = 2 and 4 positions were used to check the test data of FBG sensors at positions *h*/B = 1, 3 and 5. For the model pile TP2 pile, the FBG strain sensor layout can measure pile axial stress at different locations, which can be used to calculate the relative distance from the pile end *h*/B = 1, 3, and 5, which was the distribution curve of the pile-side average frictional resistance τ*_av_* of pile sections A, B, and C with soil penetration depth. The length of section A, B, and C was 0.14 m, 0.42 m, and 0.70 m for soil depths of *h* < 0.14 m, *h* < 0.42 m and *h* < 0.70 m, respectively. The pile-side average frictional resistance τ*_av_* of sections A, B, and C was calculated according to the actual depth *h* of the pile body into the soil (Figure 13).

When the burial depth was *h* < 0.14 m, *h* < 0.42 m, and *h* < 0.70 m, the FBG sensor at the corresponding positions *h*/B = 1, 4, and 6 had not been buried, so the distribution curve of the average lateral friction in sections A, B, and C along with the burial depth was the same as that at the pile side.

When the burial depth *h* exceeded 0.14 m, the average lateral friction of section A increased linearly with the burial depth. When the burial depth *h* exceeded 0.42 m and 0.70 m, the distribution law of average lateral friction in sections B and C gradually conformed to the distribution law of average lateral friction on the pile side with burial depth. 

When the burial depth was *h* > 0.14 m, *h* > 0.42 m, and *h* > 0.70 m, the average lateral friction of the pile side was less than the average lateral friction of section A, section B, and section C, of which the average lateral friction of section A was the largest, and the average lateral friction of section C was the smallest, caused by the ‘*h*/B effect’ during pile jacking. The “*h*/B effect” states that the greater the distance from the pile end, the lower the average friction resistance on the pile side will be at the same depth of entry, and there will be a ‘degradation effect’ on the average friction resistance on the pile side at the same depth. During the pile jacking process, lateral friction resistance degraded at the same horizontal depth, and the shear strength at the pile-soil interface gradually decreased with increasing the pile end depth.

The shear strength of the pile-soil interface decreased due to load transfer, and the upper soil layer was continuously disturbed during pile-soil settlement. The remolding degree of the pile-soil interface was particularly high, and the undrained shear strength significantly decreased. The reduction in pile-soil interface shear strength caused by load transfer was essentially a reduction in friction angle between the soil and pile.

### 4.4. Pile Jacking Resistance

In Figure 14, the pile jacking resistance curves of model piles TP1, TP2, and TP3 were drawn, as well as the variation curves of pile resistance *P_h_*, pile end resistance *q_b_* and pile side friction resistance *P_s_* with the burial depth. In the pile jacking process of model piles TP1, TP2, and TP3, the growth rate of pile jacking resistance changed at approximately 0.4 m. This is mainly because when the depth of burial is 0–0.4 m, the pile resistance increases linearly with burial depth. When the burial depth is greater than 0.4 m, the pile end resistance basically remains unchanged, while the pile jacking resistance increases linearly with the increasing burial depth. Therefore, the pile jacking resistance increases rapidly within the range of 0–0.4 m buried depth, and gradually slows down after exceeding 0.4 m. Whether it was the open pipe pile TP1 or closed pipe piles TP2 and TP3, the proportion of total pile lateral friction resistance in the pile jacking resistance was small, which had little influence on the change law and was also one of the reasons why the jacking resistance change law was consistent with the pile jacking resistance change law.

## 5. Conclusions

Based on FBG sensor test technology, the indoor model test of static pile was carried out, and a series of useful conclusions were obtained:

(1) The load transfer of open pipe piles under the action of the soil plug was different from that of closed pipe piles. When the soil plug tended to be occluded, the load on the open pipe pile and the closed pipe pile were similar, and the pile jacking resistance of the closed pipe pile was greater than that of the open pipe pile during penetration.

(2) The load transfer on the pile body during jacking was related to the play of the average friction resistance of the pile side, which caused the ‘*h*/B effect’ in the penetration process, that is, the ‘degradation effect’ of the average friction resistance of the pile side at the same depth.

(3) Data from the FBG sensor and pull line displacement sensor can accurately reflect the influence of soil plug formation on the pile jacking resistance of pipe pile as penetration depth increases. The data clearly show the soil plug effect on the open pipe pile as well as the soil squeeze effect and penetration characteristics on the open and closed pipe piles. FBG sensing technology can meet the testing requirements for penetration tests of pipe piles and can accurately measure the strain change of pipe piles, the axial force on pile bodies, and the penetration resistance of pile jacking.

## Figures and Tables

**Figure 1 sensors-20-01934-f001:**
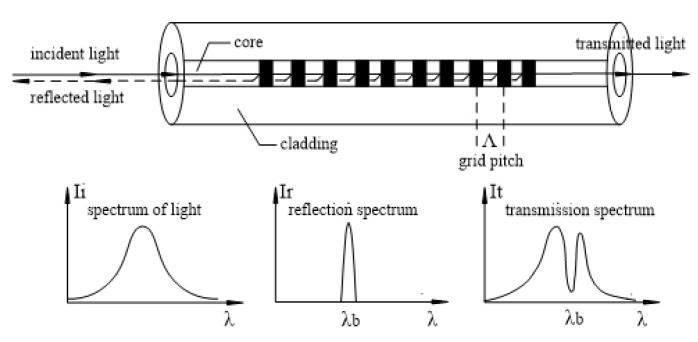
Working principle of Fiber Bragg Grating sensors. I_i_—source spectrum; I_r_—reflectance spectrum; I_t_—transmission spectrum; λ—wave length (nm); *λ_b_*—central wavelength (nm).

**Figure 2 sensors-20-01934-f002:**
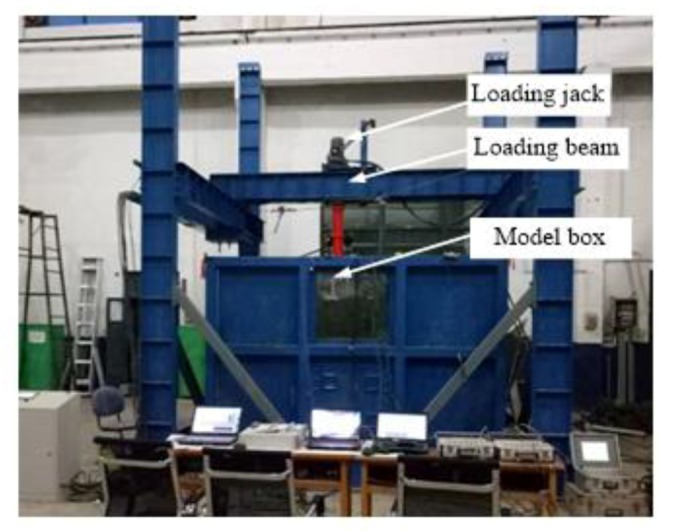
Image showing the large-scale model test system.

**Figure 3 sensors-20-01934-f003:**
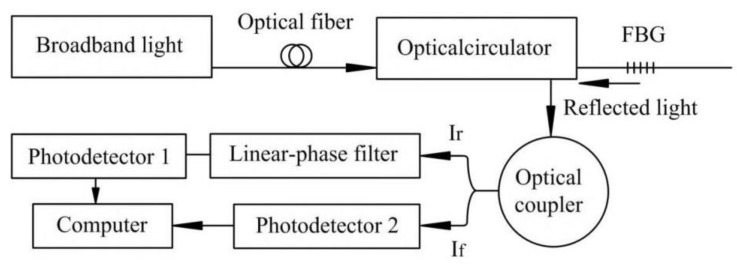
Schematic diagram of grating signal mediation.

**Figure 4 sensors-20-01934-f004:**
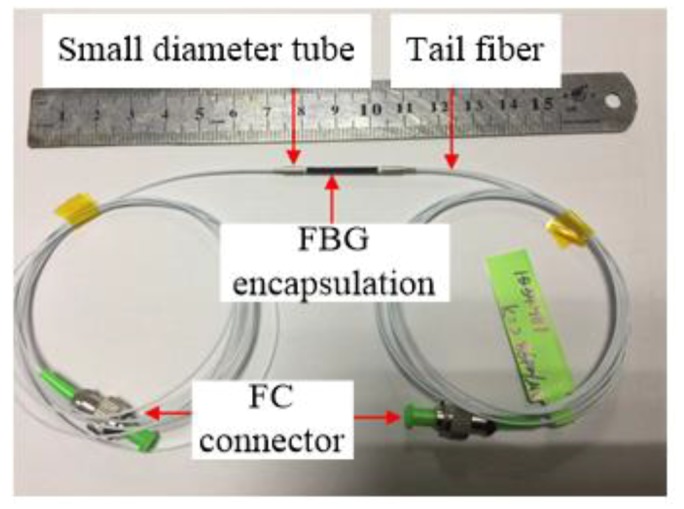
Image of the enhanced micro FBG strain sensor.

**Figure 5 sensors-20-01934-f005:**
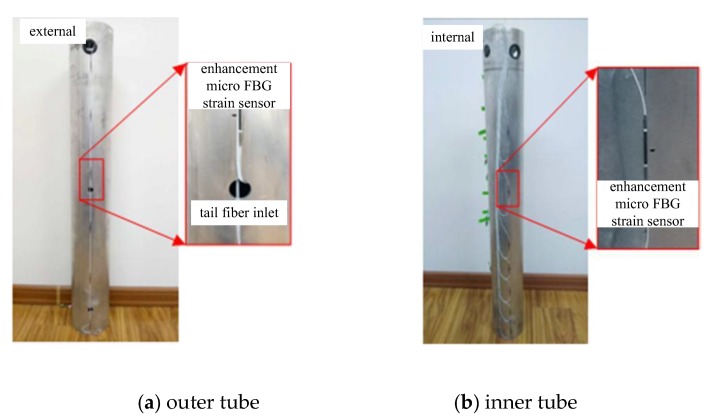
Installation of the mode pipe pile FBG strain sensor.

**Figure 6 sensors-20-01934-f006:**
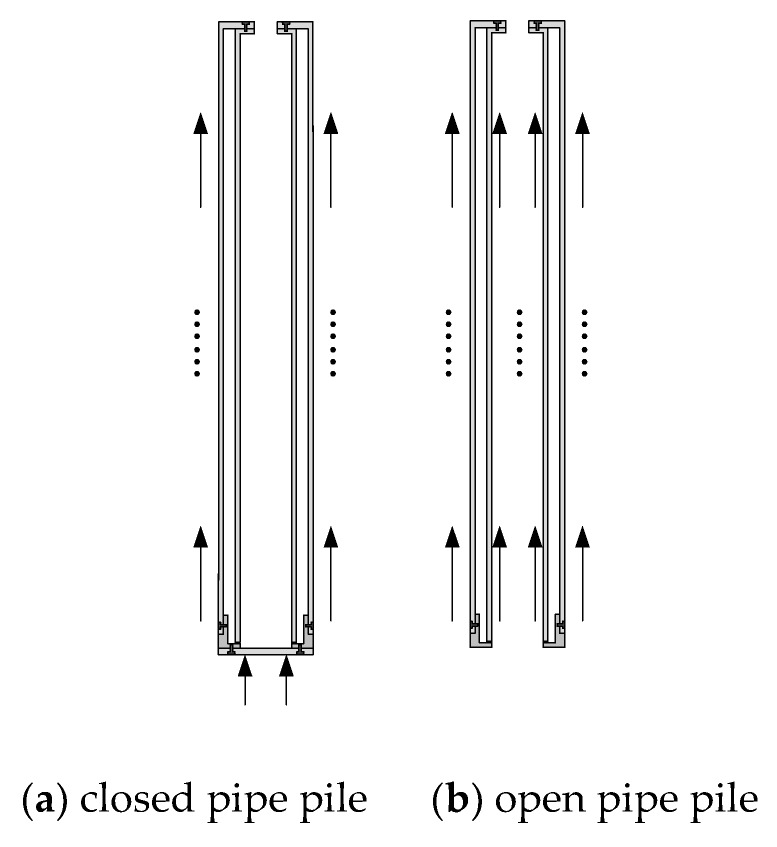
Stress mechanism of open and closed pile.

**Figure 7 sensors-20-01934-f007:**
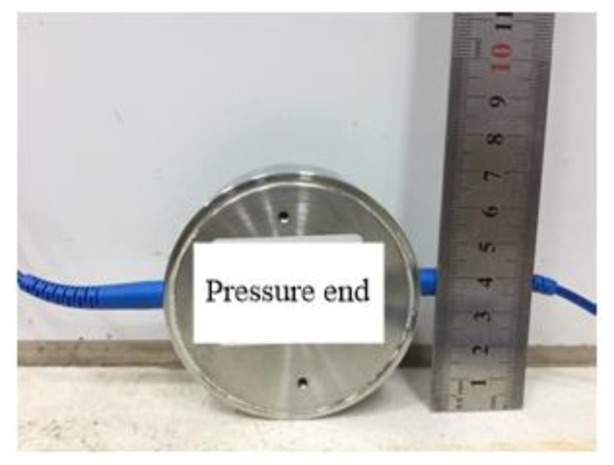
Pile top temperature self-compensating pressure sensor.

**Figure 8 sensors-20-01934-f008:**
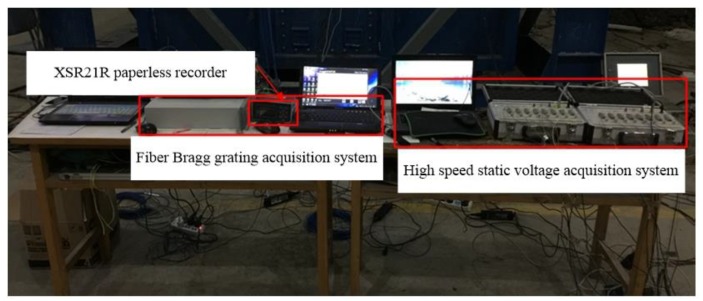
Data collection system.

**Figure 9 sensors-20-01934-f009:**
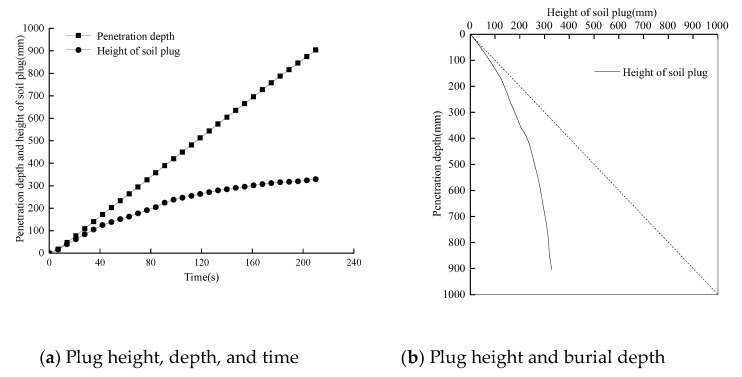
Curves of soil plug height with burial depth.

**Figure 10 sensors-20-01934-f010:**
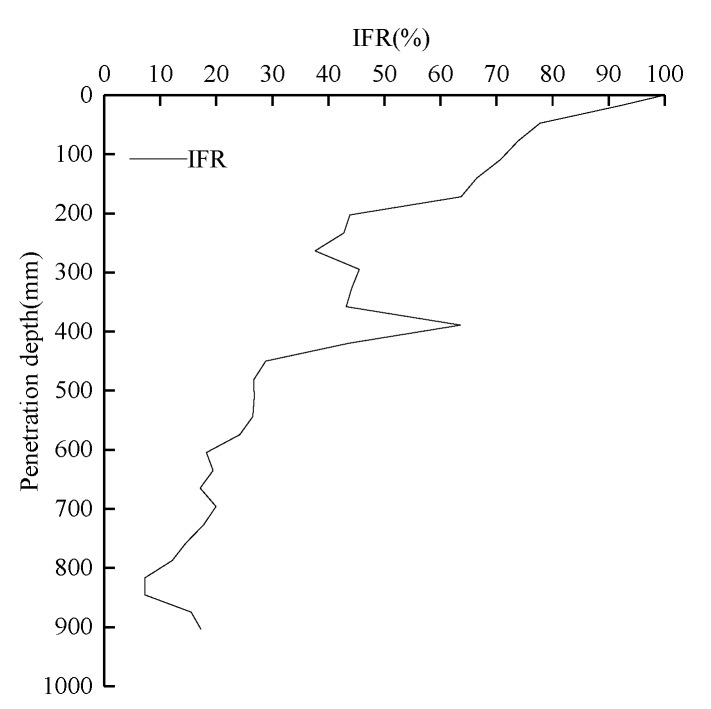
Development of Increased Frequency Reporting (*IFR*) of piles during installation.

**Figure 11 sensors-20-01934-f011:**
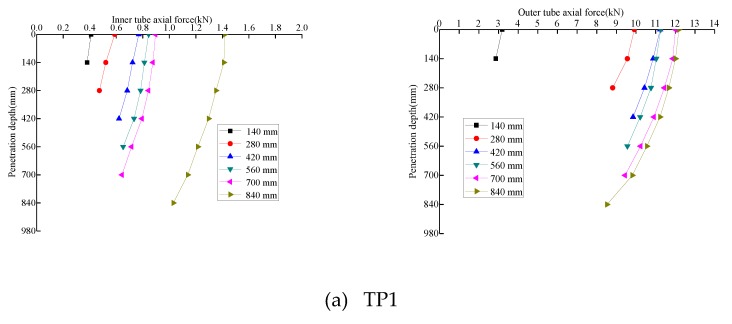
Variation of pile shaft force with depth.

**Figure 12 sensors-20-01934-f012:**
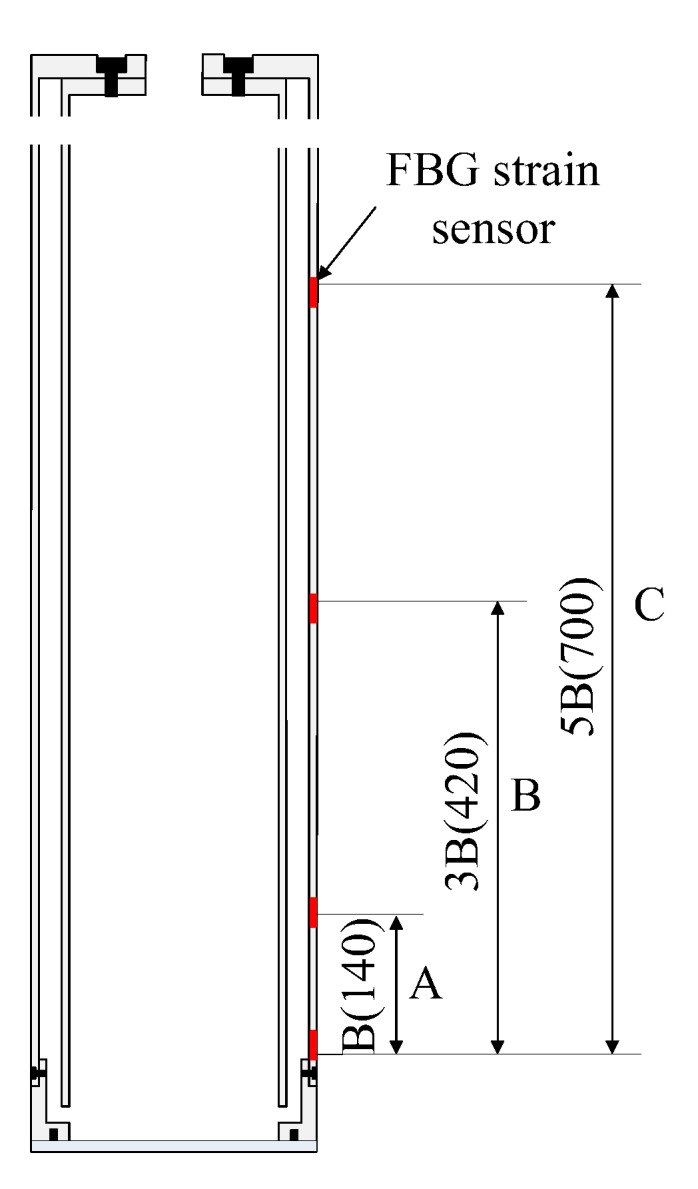
Block diagram of the pile body.

**Figure 13 sensors-20-01934-f013:**
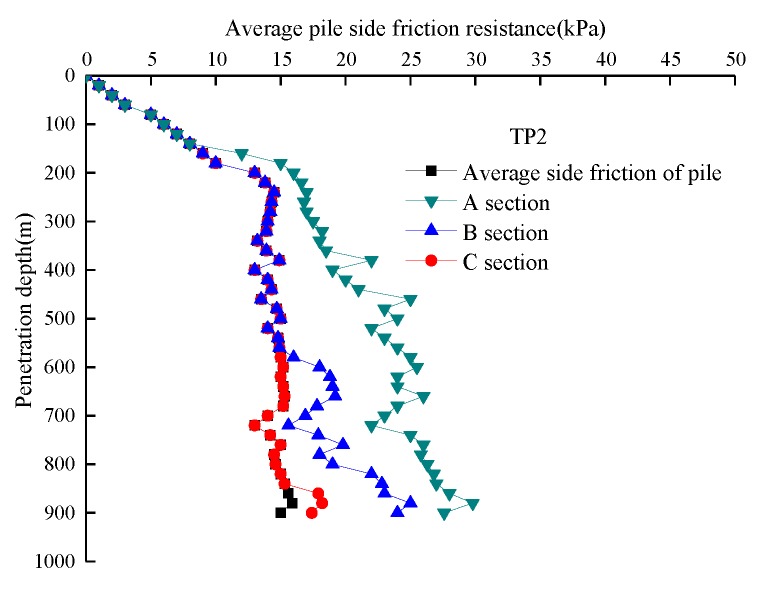
Curve of average lateral frictional resistance with penetration depth.

**Figure 14 sensors-20-01934-f014:**
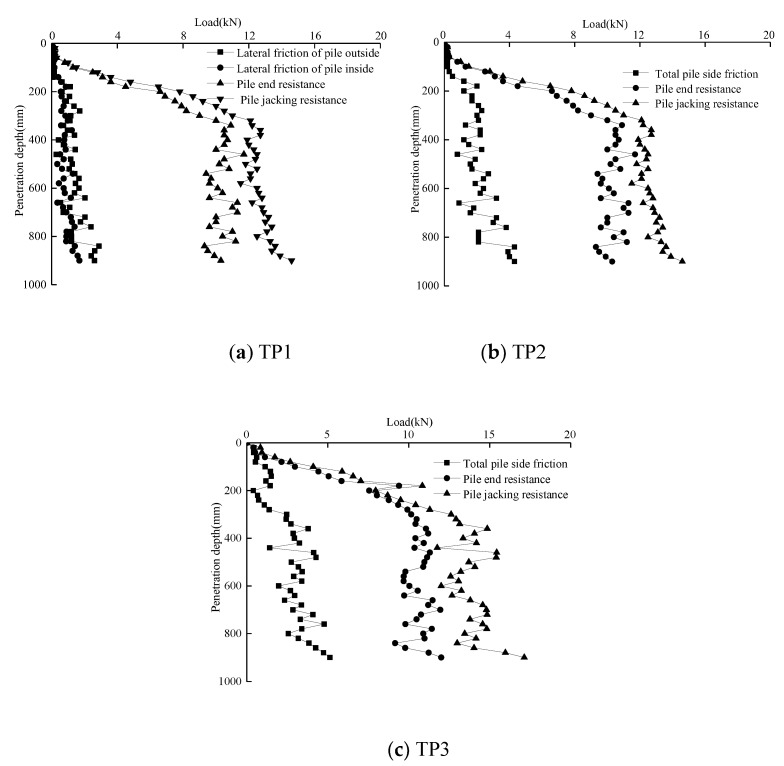
Pile jacking resistance curves of model piles TP1, TP2, and TP3.

**Table 1 sensors-20-01934-t001:** Parameters of the external pipe pile and internal tube pipe pile.

	Length (mm)	Section Radius (mm)	Section Thickness (mm)	Elastic Modulus (GPa)	Poisson’s Ratio
External tube	1000	67	3	72	0.3
Internal tube	1000	57	3	72	0.3

**Table 2 sensors-20-01934-t002:** Technical parameters of the enhanced micro FBG sensor.

3 dB Bandwidth Typical Value (nm)	Wavelength Spacing (nm)	Center Wavelength (nm)	Range (με)	Measurement Gauge (mm)	Resolution (με)
0.2	±3	1510–1590	±1500	23	1

**Table 3 sensors-20-01934-t003:** FBG parameter index demodulator parameters.

Sampling Rate (time/s)	Dynamic Range (nm)	Range (nm)	Accuracy (pm)	Resolution (pm)
1	±3	1500~1600	±2	1

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
