# Peer review of "Test and Study of Pipe Pile Penetration in Cohesive Soil Using FBG Sensing Technology"

_sensors, 2020, doi:10.3390/s20071934_

Round 1

Reviewer 1 Report

The paper shows significant work and fills the gap in knowledge. 

There are some typos and some very long sentences.

I would suggest you clearly identify the acronyms the first time you mention them, like BG, PCC, PHC, in the introduction for example.

I would suggest you make a stronger introduction to the problem, maybe adding a few lines more before you start with the literature review.

On the FBG sensor working principle (possibly section 2), I would suggest you add a couple of lines regarding how you manufacture these FBGs, where you buy the fiber, and what material the fiber is (silica, borosilicate, polymer). In this section, you mention that "the indoor test temperature change was not large" what do you mean for large? Since the FBG wavelength changes of ~13 nm/C, what is the controlling system you have used so that the temperature changes is ignored in equation 2? It would be nice to include that in the text.

The section number does not go accordingly. it is always 1. 

In the Test scheme section, why do you need to specify 3,000 mm and not 3 m? Normally, in science, the measurement is conducted at least 3 times so that you can have an average value and plus/minus the standard deviation. What is the standard deviation of this measurement? It would be nice to include that in the text.

In the Strain testing technology, what is the glue you are using? Is it always the 704 adhesive? It would be good if you can describe the adhesive and why you used this one and not other types of adhesive. What is the dimension of the "equal spacing" you are talking about in this section?

In the Temperature self-compensating pressure sensor section, what is the model of this self-compensation pressure sensor installed? It would be nice if you can include that in the paper.

In the section Data acquisition system, why do you use an FS 2200RM demodulation instrument to detect the Bragg wavelength, what are the advantages compared to a laser interrogator from Micron-Optic? Is it cheaper? It would be nice to include that in the paper.

Please consider to repeat the experiment, at least, 3 times so that you can include the standard deviation in all the graphs so that your results have a stronger meaning.

In Figure 9 could you please describe better what the legend is describing? Is it the depth that is changing from 140 to 840 mm? It would be nice to include it in the description above the figure.

Reviewer 2 Report

 This paper presents results on the jack piles embedded in soil using FBG to measure the applied forces during the embedment. The way the paper is presented is not particularly coherent and has a number of issues as outlined below:

Minor

  1. Abstract: The strain was collected using FBG sensors: the strain is not collated is measured, Need rewetting too long sentences?
  2. In the Introduction: What is "thermal BG decay "
  3. Fig 2: Fig 3 ..(The double-wall open and end model pipe pile was selected to create friction between the outer and inner walls of the pipe pile).. A schematic diagram of the application of the pipe pile can be helpful
  4. Table 2: 3dB bandwidth dB bandwidth typical value(mm) ??nm perhaps?
  5. P4: "One end of the sensor clamping sleeve was first glued to the bottom of the outer tube slot, and the other end was pre-stretched to the outer side for 2~0.8 nm, fixed with adhesive, and finally sealed with epoxy resin". This sentence does not make sense ....
  6. P5: " In order to obtain the pile jacking resistance change law during pile jacking, a temperature self-compensation pressure sensor was installed on the top of the model pile to measure the pile jacking force." If FBG on top its does not measure the ground temp. (again diagram would help)
  7. Fig 6 is not very informative

 Major :

  1. A better introduction is needed on the subject: What are "jack piles" open and close-ended. What are the specific attributes of FBG over stain gauges fro insane and what is the contribution of this method?  A schematic diagram together with the pictures like fig 10 with all the details of the pile would help ...  
  2. The terms and data without adequate definitions or explanations e.g. graphs in fig 9  in and 10 P6,7.8  need better explanations and equations 5, 6,  need also references.
  3. This paper presents without cohesion a sequence of results without adequate explanation of the mechanism and the pile work and significance and the sensor technology.
  4. There are no calibrations of the FBG with strain or temperatures.
  5. English also needs improvements, as it has mistakes, which makes it difficult to understand.
  6. It looks an interesting paper but it cannot be published as-is. It requires major improvements to its cohesion and a better explanation of the results.

Reviewer 3 Report

1. Equation(3) and (4) are obtained under assumption that temperature change is ignored. Althought the object of the study is to examine the applicability of fiber Bragg grating (FBG) sensing technology in
the static penetration of pipe piles, Is this assumption appropriate?

2.,For figures in the paper such as Figure 4,the text and value in the picture are not clear.
  3.In Section 3, a flow chart of sensor, measurement, acquisition, storage of signals for the whole test is suggest to be given,so that readers can understand the author's test clearly and intuitively.
4.It is suggested that the author modify the Abstract and Conclusions in detail. Also, there are also many errors in the sequence labels.

Round 2

Reviewer 1 Report

Well Done!

Author Response

Thanks for your advice!

Reviewer 2 Report

The Authors have indeed improved the presentation of their results and I believe it is ready to be published in its present form.

One small point : it is not clear who is the corresponding Author 

Author Response

Thank you very much for your comments! When the article was first modified, we updated the authors and changed the corresponding author to Mingyi Zhang.

Reviewer 3 Report

1.The writing of this paper is strongly recommended to revised. There are example such as:τav in equation (6) and line 235 is different,the format of p11 and p12 in line 102 is wrong,λb in lin 89 and equation (2) is different.
High academic rigor is recommended.

2.Regarding the question in last revision "Equation(3) and (4) are obtained under assumption that temperature change is ignored. Althought the object of the study is to examine the applicability of fiber Bragg grating (FBG) sensing technology in the static penetration of pipe piles, Is this assumption appropriate?",sufficient documentary evidences are suggested to provide.

Author Response

The second change is marked as a revision to distinguish it from the first.
